# Edge physics at the deconfined transition
# between a quantum spin Hall insulator and a superconductor

Ruochen Ma,[1, 2] Liujun Zou,[1] and Chong Wang[1]

[1]*Perimeter Institute for Theoretical Physics, Waterloo, Ontario, Canada N2L 2Y5*
[2]*Department of Physics and Astronomy, University of Waterloo, Waterloo, Ontario, Canada N2L 3G1*

We study the edge physics of the deconfined quantum phase transition (DQCP) between a spontaneous quantum spin Hall (QSH) insulator and a spin-singlet superconductor (SC). Although the bulk of this transition is in the same universality class as the paradigmatic deconfined Neel to valence-bond-solid transition, the boundary physics has a richer structure due to proximity to a quantum spin Hall state. We use the parton trick to write down an effective field theory for the QSH-SC transition in the presence of a boundary. We calculate various edge properties in an $N \to \infty$ limit. We show that the boundary Luttinger liquid in the QSH state survives at the phase transition, but only as "fractional" degrees of freedom that carry charge but not spin. The physical fermion remains gapless on the edge at the critical point, with a *universal jump* in the fermion scaling dimension as the system approaches the transition from the QSH side. The critical point could be viewed as a *gapless* analogue of the quantum spin Hall state but with the full $SU(2)$ spin rotation symmetry, which cannot be realized if the bulk is gapped.

## I. Introduction

The deconfined quantum critical point (DQCP) is a prototypical example of quantum phase transitions beyond Landau's paradigm [1, 2]. While DQCP has been often discussed in the context of quantum magnetism, as a transition between a Neel antiferromagnet and a valence-bond-solid (VBS), the same bulk universality class describes the transition, in a fermion system, from a spontaneous quantum spin Hall (QSH) insulator to a spin-singlet s-wave superconductor (SC) [3, 4]. In the fermionic realization, the QSH insulator spontaneously breaks the spin $SU(2)$ symmetry down to a $U(1)_z$ while preserving the charge $U(1)_c$ conservation; the superconductor in contrary breaks the charge $U(1)_c$ while preserving the full $SU(2)$.

The fermionic realization of DQCP, as a QSH-SC transition, has some features not present in the Neel-VBS transition. The QSH insulator [5, 6], being an early example of symmetry-protected topological (SPT) phase [7, 8], has gapless fermion edge modes protected jointly by the unbroken charge $U(1)$ and spin $U(1)_z$ (or time-reversal $\mathbb{Z}_2^T$) symmetries. A natural question is: how would this aspect of SPT physics affect the phase transition, especially when the system has a boundary?

The broader topic of the interplay between SPT physics and bulk criticality has attracted some attention in recent years [9–22]. In particular, for the QSH-SC deconfined transition, Ref. [16] argued, based on the notion of emergent anomaly, that the physical fermion must be gapless on the edge even though it is gapped in the bulk – recall that all the gapless degrees of freedom in the bulk are bosonic, carrying integer spins (the QSH order parameter) and even electric charges (the Cooper pair). Heuristically the gaplessness of the edge fermions can be understood as a consequence of the gapless edge states in the QSH phase, together with the

continuity of the transition. The emergent anomaly, however, further predicts a specific form of topological response that is an analogue of the quantum spin Hall effect, but with the full $SU(2)$ spin rotation symmetry – this is well known to be impossible for a gapped bulk system.

In this work, we explicitly write down an effective field theory for the QSH-SC deconfined transition, in the presence of a boundary. The field theory is a gauge theory constructed using the parton trick [23]. Our field theory takes the usual form of $CP^1$ gauge theory in the bulk, containing a dynamical $U(1)$ gauge field coupled with a critical Higgs field that carries spin-1/2. On the boundary, however, there is an additional pair of helical fermions which couples to the dynamical $U(1)$ gauge field. These helical fermions, carrying electric charge but not spin, are the remnants of the QSH edge states at the critical point. We discuss various kinematic aspects of this theory, including details of the parton construction and the choice of boundary conditions, in Sec. II.

In Sec. III, using the large-$N$ technique (with the spin rotation symmetry generalized to $SU(N)$), the boundary critical behaviours of this quantum phase transition are analyzed. As we will demonstrate, the boundary helical fermions dynamically decouple from the critical bulk at large $N$, making the problem tractable. Since the symmetry "fractionalizes" at the DQCP, the expressions of various symmetry order parameters at the edge not only involve the helical edge fermions, but are also dressed by the critical degrees of freedom from the bulk. As a result, these order parameters show rather distinct scaling behaviours at the critical point from those in the familiar single-component Luttinger liquid theory [24], which describes the boundary of the neighbouring QSH insulator. We also predict a *universal jump* in the scaling dimension of the boundary fermions as the system approaches the transition from the QSH side. For $N$

small ($N = 2$ being the directly relevant one), we argue in Sec. III that our edge theory is still stable in certain parameter regimes – in particular the helical fermions still decouple with other degrees of freedom (DOFs) as long as the scaling dimension of the Cooper pair (decided by the Luttinger parameter) is sufficiently large.

In Sec. IV we demonstrate explicitly how the edge states result in a nontrivial bulk response (also known as emergent anomaly). In particular, a $\pi$-flux of the $U(1)_c$ symmetry will trap a spin-1/2 moment in the bulk – this is analogous to the famous quantum spin Hall response, but now with the full spin $SU(2)$ symmetry. Such response is impossible if the bulk is gapped – this is the familiar statement that topological insulators require spin-orbit coupling. The bulk criticality, with the emergent anomaly, provides a route to realize this peculiar response. Another notable consequence of the emergent anomaly is that the enlarged $SO(5)$ symmetry, which emerges in the critical bulk system [25, 26], no longer exists on the boundary as long as time-reversal symmetry is unbroken.

We end with some discussions on open problems and future directions in Sec. V.

## II.   Kinematics

### A.   Parton gauge theory

The DQCP between a superconductor and spontaneous quantum spin Hall insulator is usually formulated in terms of a Wess-Zumino-Witten sigma model [3] due to its natural connection with Dirac fermions. For our purpose of studying the edge physics, it is more convenient to adapt a gauge theoretic formulation. Below we construct the gauge theory using the parton trick [23], from which the form of the edge theory will also become apparent.

The fundamental degree of freedom (DOF) in the system is the electron $c_\alpha$ (with $\alpha = 1, 2$ the spin index), which carries unit charge under the $U(1)$ charge conservation symmetry and spin-1/2 under $SU(2)$ spin rotation symmetry. To access the DQCP between the SC and QSH, we employ the following parton decomposition of the electron:

$$c = \begin{pmatrix} c_1 \\ c_2 \end{pmatrix} = Bf = \begin{pmatrix} b_1 & -b_2^\dagger \\ b_2 & b_1^\dagger \end{pmatrix} \begin{pmatrix} f_1 \\ f_2^\dagger \end{pmatrix}. \qquad (1)$$

This decomposition has an $SU(2)$ gauge redundancy, under which $B \to BU^\dagger$ and $f \to Uf$ with $U$ an $SU(2)$ matrix. We will arrange a parton mean field that breaks the $SU(2)$ gauge structure to its diagonal $U(1)$ subgroup. Consequently, the low-energy effective theory takes the following form

$$S = S_{[B,a]} + S_{[f,a]} + S_{[B,f]} + S_{[a]}, \qquad (2)$$

where $S_{[B,a]}(S_{[f,a]})$ denotes the action of $B(f)$ that is minimally coupled to $a$, an emergent $U(1)$ gauge field.

$b$ and $f$ carry opposite gauge charges, i.e., under the $U(1)$ gauge transformation,

$$b_{1,2} \to e^{-i\theta} b_{1,2} \quad f_{1,2} \to e^{i\theta} f_{1,2} \qquad (3)$$

$S_{[B,f]}$ represents the coupling between $B$ and $f$, and $S_{[a]}$ is the action of the $U(1)$ gauge field $a$.

The global $U(1)_c$ charge conservation symmetry acts on the partons, up to a gauge choice, as

$$B \to B, \quad f \to e^{i\theta} f. \qquad (4)$$

Under the $SU(2)$ spin rotational symmetry,

$$B \to VB, \quad f \to f, \qquad (5)$$

where $V$ is an $SU(2)$ matrix. Under time reversal symmetry $\mathcal{T}$, one has

$$B \to B^*, \quad f \to i\sigma^2 f, \quad i \to -i. \qquad (6)$$

Note that $\mathcal{T}$ is an anti-unitary symmetry. One can check that $\mathcal{T}$ takes $c \to i\sigma^2 c$, as $B$ has a structure of an $SU(2) \cong USp(2)$ matrix. We choose the boson $B$ to be a Kramers singlet, which will be important in the following discussion.

Next, we put the $f_1$ fermion into a Chern band with Chern number $C = +1$, and $f_2$ into a Chern band with $C = -1$. One can see from the above symmetry implementation that $f_1$ couples to the gauge field $-(a + A)$ and $f_2$ couples to $-(a - A)$, where $A$ is the probe gauge field of the global $U(1)$ charge conservation symmetry. Integrating out the fermions converts $S_{[f,a]}$ into a Chern-Simons response

$$\frac{1}{4\pi}(a+A)d(a+A) - \frac{1}{4\pi}(a-A)d(a-A) = \frac{1}{\pi} A da. \quad (7)$$

The coefficient of the Chern-Simons response is such that a strength-1 monopole operator $\mathcal{M}_a$, which creates $2\pi$ flux of $a$, carries physical electric charge $Q = 2$ - it will be identified with a spin-singlet Cooper pair of electrons [1].

Now we are ready to apply this gauge theory to describe different quantum phases. The bosons, $B$, can be condensed or gapped out by tuning the boson mass in $S_{[b_\alpha,a]}$. If the $b_\alpha$ fields are gapped, we can simply integrate them out, which induces a Maxwell dynamics for the gauge field $a$. Then the monopoles proliferate, i.e., $\langle \mathcal{M}_a \rangle \neq 0$, leading to a short-range entangled phase [27], which should be identified as a spin-singlet SC, due to the quantum numbers of the monopole operator.

---

[1] The existence of local fermionic degrees of freedom with spin-1/2 and $Q = 1$ (the electrons), which carry half-charge and half-spin of a strength-1 monopole, guarantees the global symmetries can be realized non-anomalously. This is in contrast to the standard DQCP in the context of quantum magnets. See Sec. IV and Appendix B for more details.

On the other hand, a QSH results if the $b_\alpha$ fields are condensed. To see it, suppose $b_1$ is condensed with a real-valued expectation $\langle b_1 \rangle \neq 0$ and $\langle b_2 \rangle = 0$. [2] In this case the $SU(2)$ spin rotational symmetry is broken to $U(1)_z$, the rotation along $z$-axis, while the time reversal symmetry is intact (which is due to the fact that $B$ is a Kramers singlet). The gauge field $a_\mu$ is then Higgsed and the monopole is no longer relevant. Moreover, the $f$ fermions can be identified as an electronic quasiparticle with a renormalized quasiparticle residue $Z \sim |\langle b_\uparrow \rangle|^2$, according to the relation in Eq. (1). Due to the Chern numbers of the $f$ fermions, the resulting state is simply a QSH, with a gapped bulk and a gapless left-moving (right-moving) spin-up (spin-down) mode on the edge.

The story gets more interesting right at the SC-QSH critical point, where the $b_\alpha$ bosons are critical. The bulk physics of this transition has already been investigated [3, 4], with a critical theory given by the following $CP^1$ model

$$S_{\rm bulk} = \int d^3x \left[ \sum_{\alpha=1,2} |D_\mu b_\alpha|^2 + \frac{\lambda}{4}(|b_1|^2 + |b_2|^2)^2 \right] \quad (8)$$

with $D_\mu = \partial_\mu - ia_\mu$ the covariant derivative. This is also the critical theory proposed for the DQCP of spin-$1/2$ antiferromagnet on a square lattice. The bulk critical properties of this theory has been studied extensively over the past two decades. A partial sampling of numerical works include Ref. [25, 28–31]. One subtlety that emerged recently is the possibility that the transition may be very weakly first order due to fixed points collision [26, 31–35] (some more direct numerical evidences also emerged recently [36–38]). This subtlety does not arise in local correlation functions as long as the temperature is not too low, so in this work we will ignore this issue and treat the above gauge theory as describing a truly continuous transition.

We now proceed to describe the edge physics, which has been largely unexplored so far. The bulk + boundary system at the critical point is described by

$$S = S_{\rm bulk} + \int_{\rm edge} d^2x \Sigma_{p=L,R} \psi_p^\dagger (\partial_\tau + ia_\tau)\psi_p + \mathcal{H}_{\rm edge}$$
$$+ \mathcal{H}_{\rm b-e} \quad (9)$$

where $\psi_{L,R}$ are the left- and right-moving edge modes resulted from the $f$ fermions, with the edge Hamiltonian

$$\mathcal{H}_{edge} = \psi_L^\dagger(-i\partial_x + (a+A)_x)\psi_L$$
$$- \psi_R^\dagger(-i\partial_x + (a-A)_x)\psi_R + ... \quad (10)$$

---

[2] One can always bring the condensation pattern into this form by a spin rotation in Eq. (5).

where ... includes other edge couplings (such as four-fermion interactions) allowed by symmetry and gauge invariance. Note although $\psi_{L,R}$ lives on the edge, $a$ in this equation lives on the edge and also the bulk. $\mathcal{H}_{\rm b-e}$ denotes other coupling between the bulk and the edge – we shall discuss these additional couplings in more detail in Sec. III.

## B. Boundary conditions

We now consider the appropriate boundary conditions for the bulk fields $b_\alpha$ and $a_\mu$. We denote the boundary as $z = 0$ separating the bulk $z < 0$ from the vacuum $z > 0$, and the two other coordinates parrallel to the edge spacetime are denoted $x, \tau$.

For the scalar field $b_\alpha$ we will only consider the Dirichlet boundary condition $b_\alpha|_{z=0} = 0$, which is also known as the "ordinary" boundary condition, where the scalar fields on the boundary and in the bulk condense simultaneously as the bulk mass changes sign. This boundary condition is particularly appropriate in the large-$N$ generalizations to be considered in Sec. III. Other more exotic possibilities may exist for relatively small $N$ [39], but are beyond the scope of this work.

For the gauge field $a_\mu$, it turns out that we should also adopt the Dirichlet boundary condition $a|_{z=0} = 0$ (up to a gauge transformation). The reason is again due to the large-$N$ generalization of the theory to be considered in Sec. III. In this generalization, when the number of flavors of the matter fields $b_\alpha$ is large, for all spacetime dimensions $d < 4$, the effective photon dynamics at low energies is entirely induced by the bosonic matter $b_\alpha$ [27, 40]. The Dirichlet boundary condition $b_\alpha|_{z=0} = 0$ then immediately leads to the Dirichlet boundary condition of the gauge field $a_\mu$. Physically, the matter fields $b_\alpha$ and $\boldsymbol{e}$, the electric field of $a$, change continuously at the system boundary when the Dirichlet boundary conditions are imposed. We will come back to this choice of boundary conditions in Sec. III. An additional remark is that this discussion applies to the case where the system is a half infinite plane, and it should be slightly modified if the system is, say, a disc (see Appendix A).

The vanishing of electric field $\boldsymbol{e}|_{z=0} = 0$ at the edge also means that when we put test gauge charges on the boundary, the electric field line emitted from the gauge charge cannot be restricted to the boundary (since the field strength must vanish there) and must penetrate into the bulk. So the usual mechanism of $1d$ confinement, due to the electric field line restricted to $1d$ with a finite tension, does not apply on the boundary, and we expect the boundary gauge charges to be deconfined. This is just the boundary manifestation of the familiar deconfinement of $U(1)$ gauge theory with a conserved flux-current.

## III. Edge dynamics

Let us first make some general remarks on the edge physics. In the SC, there is no nontrivial edge mode. In the QSH, the edge has a $U(1)_c \times U(1)_z$ symmetry, which cannot be broken spontaneously due to the Mermin-Wagner theorem, and which carries the standard axial 't Hooft anomaly. This implies that the edge state in the QSH cannot be gapped. Quite generically, it can be described by a Luttinger liquid.

The bosonized description of a Luttinger liquid reads [24],

$$S_{\rm LL} = \frac{1}{2\pi} \int_{x,\tau} [\frac{(\partial_x \phi)^2}{K} + K(\partial_x \theta)^2] + \frac{i}{\pi} \int_{x,\tau} \partial_x \theta \partial_\tau \phi, \tag{11}$$

where $K$ is the effective Luttinger liquid parameter, and $\phi$ and $\theta$ are two $\pi$-periodic compact bosons, satisfying

$$[\partial_x \phi(x), \theta(x')] = [\partial_x \theta(x), \phi(x')] = i\pi\delta(x - x'). \tag{12}$$

Various microscopic operators are expressed as vertex operators, e.g., $\Psi_R \sim e^{-i\theta + i\phi}$, $\Psi_L \sim e^{-i\theta - i\phi}$ and $S^+ \sim \Psi_L^\dagger \Psi_R \sim e^{i2\phi}$, where $\Psi_{L,R}$ are the left- and right-moving edge *electronic* operators (which carry no charge under $a$), and $S^+$ is the $XY$ spin order parameter[3]. Another interesting operator is the Cooper pair, $\Psi_L \Psi_R \sim e^{-i2\theta}$. The charge density and the spin density are expressed in terms of the bosons as $\rho_c = \frac{1}{\pi}\partial_x \phi$ and $\rho_s = \frac{1}{\pi}\partial_x \theta$, respectively. The time reversal symmetry $\mathcal{T}$ acts as

$$\theta \to -\theta, \quad \phi \to \phi + \pi/2, \tag{13}$$

which is chosen such that the electric charge is even under $\mathcal{T}$ while the spin is odd under $\mathcal{T}$.

The scaling dimensions of the vertex operators depend on the Luttinger parameter $K$ [24, 41]. The most important operators for us are the electrons $e^{-i\theta \pm i\phi}$ with $\Delta_e = (K + 1/K)/4$, the $XY$ spin operator $e^{i2\phi}$ with $\Delta_s = K$, and the Cooper pair $e^{-i2\theta}$ with $\Delta_c = \frac{1}{K}$. Certain combinations of these dimensions are universal (independent of $K$), for example

$$\Delta_s \Delta_c = 1. \tag{14}$$

Now we tune the bulk to the DQCP between the SC and QSH. From considerations based on emergent anomalies, nontrivial edge states still exist at the DQCP [15]. In particular, the fermions, which are

———

[3] The boundary conditions on $\theta$ and $\phi$ are coupled, such that the expressions of the chiral fermions are compatible with the periodicity of $\theta$ and $\phi$. Also do not confuse the physical edge modes $\Psi_{L,R}$ with the edge modes of the partons $\psi_{L,R}$. In the quantum spin Hall phase, we have $\Psi_L \propto \psi_L$ and $\Psi_R \propto \psi_R^\dagger$.

gapped in the bulk throughout the transition, are required to be gapless on the boundary. However, the detailed edge physics remains unknown. For example, one may ask how the scaling dimensions of various operators will differ from a simple Luttinger liquid.

The edge behaviour of the DQCP can be systematically studied in certain large-$N$ generalization of the action in Eq. (9). We start by ignoring the gapless fermions and focusing on the sector of $b_\alpha$ and $a$. Letting the bulk gauge field $a_\mu$ be coupled to $N$ species of critical bosons, $S_{\rm bulk}$ becomes a $\rm CP^{N-1}$ model

$$S_{\rm bulk}^{(N)} = \int_{bulk} d^3x \sum_{j=1}^{N/2} \sum_{\alpha=1,2} |D_\mu b_{j,\alpha}|^2 + \frac{\lambda}{2N}(\sum_{j,\alpha} |b_{j,\alpha}|^2)^2. \tag{15}$$

The explicit $N$-dependence in the generalized action has been chosen to lead to a consistent large-$N$ limit with $\lambda \sim \mathcal{O}(1)$. The global symmetry of the model is

$$(\mathrm{PSU}(N) \times U(1)_{top}) \rtimes Z_2^T, \tag{16}$$

where $U(1)_{top}$ denotes the topological $U(1)$ symmetry, whose conserved current is the flux of $a_\mu$, $J^\mu = \frac{1}{2\pi}\varepsilon^{\mu\nu\rho}\partial_\nu a_\rho$. As discussed in the previous section, the $U(1)_{top}$ corresponds to the microscopic $U(1)$ charge conservation symmetry. $Z_2^T$ is the time reversal, which acts as

$$\mathcal{T} : b_i \to b_i^*, \quad a_\mu \to a_\mu. \tag{17}$$

The $\mathcal{T}$ action above is indicated for the spatial components. The time component will transform with opposite sign. The $PSU(N)$ symmetry reduces to the spin rotational symmetry at $N = 2$. We will use symmetry representations to organize the operators in the large-$N$ discussion below.

The standard technique to study the theory in Eq. (15) at large-$N$ is to introduce a Hubbard-Stratonovich auxiliary field $\sigma$, such that

$$S_{\rm bulk}^{(N)} = \int_{bulk} d^d x \sum_{i=1}^N |D_\mu b_i|^2 + \sigma|b_i|^2 - \frac{N}{2\lambda}\sigma^2, \tag{18}$$

where the indices $(j, \alpha)$ are collected into a single index $i$. Integrating out the $\sigma$ field, one reproduces the original action. Eq. (18) is now Gaussian in $b_i$, and the integral over the field $b_i$ can be performed to yield:

$$S_{[\sigma, a_\mu]} = N \mathrm{Tr} \log(-D_\mu^2 + \sigma) - \frac{N}{2\lambda}\sigma^2. \tag{19}$$

This effective action is proportional to $N$, so the fluctuations of $\sigma$ and $a_\mu$ are suppressed in the large-$N$ limit, and the behaviour of these fields are dominated by the saddle point equations. In particular, the saddle point equation for the gauge field reads

$$\mathrm{Tr}[\frac{(\partial_\mu - ia_\mu)}{-D_\nu D^\nu + \sigma}] = 0. \tag{20}$$

By translational and rotational symmetry in the $x\tau$ plane, $a_\mu = 0$ is always a solution for $\mu = x, \tau$. The equation for $a_z$ is much more non-trivial due to the presence of a boundary. However, one can utilize the gauge freedom to set $a_z = 0$. We thus conclude that, similar as the case of infinite systems, the gauge field plays no role in the saddle point equations, which then reduce to the saddle point equations of the $O(2N)$ non-linear sigma model (NLSM) in the semi-infinite plane [40]. As we expand the functional determinant in Eq. (19) around the saddle point, one can see that the effective photon propagator is proportional to $\frac{1}{N}$, so the gauge fluctuations are negligible at large $N$ [27]. Therefore, the scaling behaviour of the model should be that of the $O(2N)$ NLSM in the $N \to \infty$ limit.

Let us now turn our attention to the boundary critical behaviour of the $O(2N)$ NLSM. For bulk spacetime dimension $d > 3$, there are three possible universality classes: the ordinary, the extraordinary and the special [42]. The ordinary universality class denotes the case where the boundary remains disordered and gapped throughout the bulk disordered phase. For the extraordinary fixed point, the bulk undergoes a symmetry breaking phase transition in the presence of an ordered boundary. The multi-critical point between the ordinary universality class and the extraordinary universality class is known as the special transition. In the physical dimension $d = 3$, a modified version of the extraordinary universality class could survive for $N$ smaller than some critical value [39]. However at $N = \infty$ one finds only an ordinary universality class [43]. This can be understood physically by noting that for spacetime dimension $d = 3$, the boundary does not order when the bulk is disordered, due to the Mermin-Wagner constraint. The appropriate boundary condition for the ordinary transition is the Dirichlet boundary condition $b_\alpha|_{z=0} = 0$, which justifies the choice made in Sec. II B.

At the critical point, the form of the correlation function on a semi-infinite plane is fixed by conformal invariance. We have [44]

$$\langle \phi^i(x,z)\phi^j(x',z')\rangle = \frac{\delta^{ij}}{(zz')^{\Delta_\phi}}\Phi(v), \quad v = \frac{\rho^2 + z^2 + z'^2}{2zz'},$$
(21)

with $\rho = |x - x'|$ the separation along the $x\tau$ plane and $\Phi$ a universal function. $\Delta_\phi$ is the bulk scaling dimension of the $\phi$ field, the $O(2N)$ vector, which is related to our original $CP^{N-1}$ field via $b_j = \phi_{2j-1} + i\phi_{2j}$. RG calculations and exact results suggest that for $\rho \to \infty$ with $z, z'$ fixed, $\Phi$ decays as $\rho^{-2\Delta_\phi^s}$ [42, 43], where $\Delta_\phi^s$ can be defined as a surface critical exponent. We thus expect an operator product expansion near the boundary

$$\phi_i(x,z) \sim \mu_\phi z^{\Delta_\phi^s - \Delta_\phi}\hat{\phi}_i(x) + ..., \quad (22)$$

where $\mu_\phi$ is a universal constant only depending on the normalization of the surface operator $\hat{\phi}_i$. (We denote

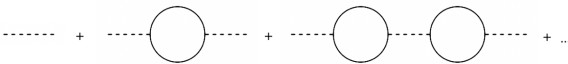

FIG. 1. The induced propagator of the $\sigma$ field. The dashed line stands for the $\sigma$ propagator at tree level. The solid line stands for the exact propagator of $b_i$ at the $N \to \infty$ limit.

boundary version of the scaling operators with a hat.) The operator $\hat{\phi}_i$ can be viewed as the leading representation of the scaling operators with a certain symmetry quantum number at the surface.

For the ordinary universality class in bulk dimension $d$, the exact result of the correlation function at $N \to \infty$ is obtained by Bray and Moore [43]:

$$\langle \phi^i(x,z)\phi^j(x',z')\rangle$$
$$\sim \delta^{ij}\left(\frac{zz'}{[\rho^2 + (z-z')^2][\rho^2 + (z+z')^2]}\right)^{(d-2)/2}. \quad (23)$$

Therefore, at $d = 3$ the scaling dimension of the bulk field $b_i$ is $[b_i] = \Delta_\phi = 1/2$, and the scaling dimension of the boundary field $\hat{b}_i$ is $[\hat{b}_i] = \Delta_\phi^s = 1$. By the equation of motion of the Hubbard-Stratonovich field $\sigma$, we see that $\sigma$ plays the role of the $PSU(N)$ singlet operator in any correlation functions. The induced propagator of $\sigma$ is obtained from summing the geometric series of bubbles in Fig. (1), which gives [45]

$$\langle \sigma(x,z)\sigma(x',z')\rangle \propto \frac{(zz')^{-2}}{N}\frac{v}{(v^2-1)^2}. \quad (24)$$

We thus read off the scaling dimensions[4] $\Delta_\sigma = 2$ and $\Delta_{\hat{\sigma}} = 3$, using the behaviour of the above correlation function in the limit $\rho \to \infty$ with $z, z'$ fixed. Since the edge singlet operator $\hat{\sigma}$ has scaling dimension $\Delta_{\hat{\sigma}} > 2$, there is no relevant perturbations at the boundary that respect the $PSU(N)$ symmetry. Another important surface operator is the boundary descendent of the gauge field $\hat{a}_\mu$. The effective photon propagator at the leading order, namely $O(1/N)$, is obtained by summing similar geometric series as that in Fig. (1) [46]. By a simple dimensional analysis, one can obtain the boundary scaling dimension of the gauge field is $\Delta_{\hat{a}} = 2$.

Now we include the gapless boundary fermions. Again we bosonize the fermion modes and write it as a Luttinger liquid Eq. (11), with $\psi_R \sim e^{i\theta - i\phi}$ and $\psi_L \sim e^{-i\theta - i\phi}$. The Luttinger liquid has a minimal coupling to the boundary gauge field, which can be written as $\frac{1}{2\pi}\hat{a}_\mu\epsilon^{\mu\nu}\partial_\nu\theta$. Since the gauge field has a boundary scaling dimension $\Delta_{\hat{a}} > 1$, this coupling is

————

[4] This agrees with the expectation that $\hat{\sigma}$ should also be the boundary descendent of the bulk stress tensor [42], which always has dimension $d = 3$.

irrelevant and has no consequence in terms of dynamics (the gauge field still imposes a global constraint of vanishing gauge charge). There are additional couplings between the Luttinger liquid and other degrees of freedom, but they all turn out to have scaling dimension larger than $d - 1 = 2$ and are therefore irrelevant (at least when generalized to large $N$): (a) the operator $e^{2i\phi}$ carries gauge charge 2 and is time-reversal odd, so it couples to $\epsilon^{\alpha\beta} b_\alpha \partial_t b_\beta$ which is irrelevant on the edge; (b) the operator $e^{2i\theta}$ is the physical Cooper pair and therefore couples to the monopole operator $\mathcal{M}_a$ of the bulk $U(1)$ gauge field. At large $N$ the monopole operator has scaling dimension $\sim O(N)$ so this coupling is again irrelevant [47, 48]; (c) the operator $\partial_x\theta$ is odd under time-reversal, and could couple with the gauge field $e_z = \partial_0 a_z - \partial_z a_0$, but since the gauge flux is a conserved current, $[e_z] = 2$ in the bulk so the boundary coupling is irrelevant; (d) the operator $\partial_x\phi$ transforms trivially under all microscopic symmetries (this is just a chemical potential term), so the leading coupling involving this operator is with either $\hat{\sigma}$ or the boundary descendent of the gauge magnetic field $\partial_x a_y - \partial_y a_x$, and in either case the coupling will be irrelevant.

Notice that all the above couplings are already irrelevant in the $N \to \infty$ limit, so including further $O(1/N)$ corrections will not immediately change the picture. We therefore conclude that, at least when $N$ is sufficiently large, the DOF in the CP$^{N-1}$ model (either living in the bulk or edge) and the edge $\psi_{L/R}$ fermions decouple in the RG sense. So the edge $\psi$ fermions behave the same as ordinary Luttinger liquids, with a Luttinger liquid parameter $K$ that is exactly marginal. For smaller values of $N$, most of the above edge couplings will still be irrelevant. The only exception is the coupling $e^{2i\theta}\mathcal{M}_a$, since $[\mathcal{M}_a]$ can become quite small for small $N$. In this case the coupling is irrelevant only if $[e^{2i\theta}] = 1/K$ is sufficiently large, i.e. if $K$ is smaller than some critical value $K_c$. Practically, this requires the Cooper pair to be "heavy" on the edge, and can be achieved, if necessary, by turning on repulsive interactions on the edge. Therefore for small $N$, what we have found is a boundary fixed line that exists in certain parameter regimes. If $K > K_c$, the Luttinger liquid no longer decouples, and the theory flows to strong coupling over which we do not have analytical control. In this case a natural possibility is that the Cooper pair $e^{2i\theta}$ may spontaneously condense (or logarithmically so [39]) due to the long-range couplings induced by the bulk fluctuations. The edge fermions will effectively be gapped in this scenario.

We now compute the boundary scaling dimensions of the electron, Cooper pair and spin operators in the $N \to \infty$ limit. The electron operator $\Psi_\alpha \sim b_\alpha f$, and since the Luttinger liquid of the $f$ fermions decouple with the rest of DOFs on the edge,

$$\Delta_\Psi = [b] + [f] = \frac{K}{4} + \frac{1}{4K} + 1. \qquad (25)$$

It is interesting to compare $\Delta_\Psi$ at the transition point

and in the QSH phase but close to the transition. Since the Luttinger liquid sector decouples with other DOFs on the edge, the Luttinger parameter $K$ should change smoothly as the system approaches the phase transition from the QSH phase. We therefore conclude that there is a *universal* jump in the fermion boundary scaling dimension as the system reaches the phase transition from the QSH side:

$$\delta\Delta_\Psi \equiv \Delta_\Psi^{DQCP} - \Delta_\Psi^{QSH} = 1. \qquad (26)$$

For finite $N$, we expect $\delta\Delta_\Psi$ to deviate from 1, but remains universal as long as the Luttinger liquid decouples with the other DOFs, which requires a sufficiently small $K$.

The above universal jump in the fermion boundary scaling dimension can be viewed as a hallmark of electron fractionalization at the transition. For other more conventional transitions, for example a Landau transition from the QSH to a superconductor (without restoring the full $SU(2)$ symmetry), such a universal jump is not expected even if the Luttinger liquid (from the QSH boundary modes) decouples with the bulk critical fields.

The scaling dimension of the spin operator can be obtained as follows. A large-$N$ generalization of the spin operator is $O_s \sim \sum_{j=1}^{N/2} [b_{j,1}^\dagger b_{j,2} (f_1^\dagger f_1 + f_2^\dagger f_2) + (b_{j,1}^\dagger)^2 f_1^\dagger f_2^\dagger - b_{j,2}^2 f_2 f_1]$,[5] which transforms in the vector representation of the $SO(3)$ spin rotational symmetry, and carries no electric charge. Under $\mathcal{T}$, $O_s$ has the expected transformation of the spin operator $S^+$, $O_s \to -(O_s)^\dagger$. The boundary scaling dimension of the conserved charge $(f_1^\dagger f_1 + f_2^\dagger f_2)_{\text{edge}} \sim (\psi_L^\dagger \psi_L + \psi_R^\dagger \psi_R)$ is 1, while $(f_1^\dagger f_2^\dagger)_{\text{edge}} \sim \psi_L^\dagger \psi_R^\dagger$ has dimension $K$ (similarly for $f_1 f_2$). The operator $\sum_j b_{j,1}^\dagger b_{j,2}$ transforms in the adjoint representation of the $PSU(N)$ symmetry, whose boundary scaling dimension is

$$\Delta_{\text{adj}} = 2[\hat{b}_i] = 2, \qquad (27)$$

due to the fact that the vertex correction to the scaling dimension of $PSU(N)$ adjoint operators is $O(\frac{1}{N})$. The operator $\sum_j (b_{j,1}^\dagger)^2 = \sum_j (b_{j,1}^{\dagger 2} + b_{j,2}^{\dagger 2}) + (b_{j,1}^{\dagger 2} - b_{j,2}^{\dagger 2})$ contains both $PSU(N)$ singlet and adjoint components (similarly for $\sum_j b_{j,2}^2$). Because $\Delta_{\text{adj}} < \Delta_{\hat\sigma}$, the adjoint component dominates at long distances. We conclude that the boundary scaling dimension of the $XY$ order parameter is

$$\Delta_s = 2 + \min(1, K) \qquad (28)$$

at the $N \to \infty$ limit.

---

[5] $O_s$ can be written in a more compact form in terms of the matrix notation in Eq. (1), $O_s \sim \frac{1}{2} \sum_{j=1}^{N/2} f^\dagger B_j^\dagger \sigma^+ B_j f$.

The physical Cooper pair can be expressed in terms of the edge fermions as $c_\uparrow c_\downarrow \sim \psi_R^\dagger \psi_L$. One may check it carries global electric charge 2, and is invariant under both $\mathcal{T}$ and spin rotational symmetry. In the large-$N$ limit, the edge scaling dimension of the Cooper pair is determined simply by the Luttinger parameter

$$\Delta_c = 1/K. \tag{29}$$

Unlike the Luttinger liquid on the QSH boundary, the universal relation Eq. (14) is no longer true at the critical point.

## IV.  Edge states and anomaly

We now show how our edge theory, in particular the existence of the gapless $\psi$ fermion modes, satisfies the constraints imposed by the emergent anomaly of DQCP. We remind the reader that our system has a vanishing bulk gap, while the bulk gap for single fermionic excitations remains open.

Let us first review the argument in Ref. [16], in a language that will be convenient for our purpose (we give a more detailed and formal account in Appendix B). Consider the system in a disk geometry, and turn on a global $U(1)$ flux $\int dA = \pi$. This flux is supposed to be turned on adiabatically, so that no bulk fermion can be excited, and only the gapless part of the system can respond to it. In the bulk the minimally charged gapless DOF is the Cooper pair, which sees the $\pi$-flux as effectively $2\pi$-flux. It is well known that for DQCP such a flux will trap a spin-1/2 moment of the $SO(3)$ symmetry [49, 50]. If the fermions are gapped everywhere (bulk and boundary), then below the fermion gap the $\pi$-flux is not observable from far away, and therefore should only induce a local excitation – but there is no local excitation with spin-1/2 below the fermion gap. This is a contradiction, which in turn requires the fermion gap to vanish on the boundary.

We can now check the above reasoning explicitly using our edge theory. The $U(1)$ flux is seen as a $\pm\pi$ flux by $\psi_{L/R}$ (see Eq. (10)). Following a standard integer quantum Hall (or chiral anomaly) calculation, we see that the flux should carry a unit gauge charge (under $a$) but no global $U(1)$ charge – in the bosonized language the flux corresponds to the operator $e^{i\phi}$. This is just the boundary manifestation of the QSH physics of the $f$ sector. Accordingly, when the $\pi$-flux is inserted, it will trap an excitation with gauge charge 1 from the $f$ sector, which combines with $b$ boson to yield the spin-1/2 property of the flux anticipated from our understanding of the DQCP. [6]

The above topological response (also known as emergent anomaly) is a close analogue of the celebrated quantum spin Hall effect, in which a $\pi$-flux traps an $S_z = 1/2$ (or Kramers degeneracy if the state is protected by time-reversal symmetry). This is the fully $SU(2)$ symmetric version, which is known to be impossible for gapped SPT states – this can be understood simply by noticing that if the bulk is fully gapped, an adiabatic insertion of a $\pi$-flux cannot turn the system from a spin singlet state into a doubly-degenerate spin-1/2 state. The bulk criticality is what makes this response possible. For this reason Ref. [16] calls such states "intrinsically gapless" topological phases.

Another consequence of the emergent anomaly is that while the bulk is known to have an emergent $SO(5)$ symmetry at the critical point [25, 26], at the boundary the symmetry is reduced back to the microscopic $SO(3) \times U(1)$ since the two order parameters (QSH and SC) will in general have different boundary scaling dimensions. At the anomaly level this is because the emergent $SO(5)$ anomaly [26] remains a nontrivial $\theta$-term even if fermions (in the $SO(5)$ spinor representation) are introduced, and the $\theta$-term cannot be trivialized as long as time-reversal symmetry is preserved (similar to the standard topological insulator [51]) – see Appendix B for more details. So the boundary gapless fermions can only trivialize the bulk $SO(3) \times U(1)$ anomaly, but not the larger $SO(5)$ anomaly, unless time-reversal symmetry is broken with nonzero $SO(5)$ and thermal Hall conductance. Because the bulk $SO(5)$ anomaly remains nontrivial in the presence of the time reversal symmetry, while the vacuum hosts no anomaly, the anomaly should be ill defined on the boundary since this is the interface between the bulk and the vacuum, which means that the $SO(5)$ and time reversal symmetries cannot be simultaneously preserved on the boundary. An example where $SO(5)$ is preserved but time reversal is broken on the boundary is given in Ref. [52], in which the fermions transform as $SO(5)$ spinors and the time-reversal symmetry is realized through the (anti-unitary) particle-hole symmetry in a half-filled Landau level. The boundary of this system can have the full $SO(5)$ (or more precisely $Spin(5)$), but the particle-hole symmetry of the Landau level is inevitably broken on the boundary. The edge state in this case can be formulated starting from bulk descriptions that are explicitly $SO(5)$ invariant, such as the $SU(2)$ QCD$_3$ theory in Ref. [26]. We leave the details of such edge theory to future work.

---

[6] A nontrivial question is how to observe this spin-1/2 physically, given the existence of critically fluctuating spin DOFs.

The general expectation is that the spin-1/2 moment cannot be fully screened by local gapless excitations that only carry integer spins. However the details may be complicated and deserve further investigations (for a recent work in this direction see Ref. [17]).

## V. Discussion

In this work we constructed the phase transition between an SPT phase, namely, a quantum spin Hall insulator, and a superconductor which breaks part of the protecting symmetries of the SPT phase. In contrast to the ordering transitions of SPT phases discussed in the literature [39, 53], the transition in this work separates two phases with different symmetry breaking patterns, one of which supports an SPT protected by the unbroken symmetry. The critical theory is therefore a DQCP in the bulk, instead of a conventional Ginzburg-Landau type theory. Due to the existence of the symmetry-protected edge modes, the phase transition exhibits unconventional boundary critical behaviours, which were studied using the large-$N$ technique. We calculated the edge scaling dimensions of the electron, the spin, and the Cooper pair operators at the $N \to \infty$ limit. The expressions of these order parameters may involve both the helical edge modes and the critical bosons from the bulk, as a result of symmetry fractionalization at the critical point. We discussed several measurable consequences. In particular, we predict a universal jump in the scaling dimension of the boundary fermions as the system approaches the phase transition from the QSH phase. The scaling dimensions at the transition also obey different relations than a conventional Luttinger liquid, which in principle can help distinguishing the critical point from the neighbouring quantum spin Hall state. Our results should be helpful for future numerical studies on the boundary criticality of the QSH-SC deconfined transition.

There are many potential future directions following our work. An obvious one is to compute the leading $1/N$ corrections on edge critical behaviour. Beyond the zeroth order, the emergent gauge field can mediate couplings between the edge modes and the critical bulk, which may lead to interesting new physics. A conceptually more intriguing question is whether entirely new boundary fixed point, in which the boundary helical modes no longer decouple from the other DOFs, can appear at small $N$, including the most relevant case of $N = 2$.

The insight of this paper can also be generalized to other systems with different symmetries. The key idea is to have two phases with distinct symmetry breaking patterns, one (or even both) of which can support SPT phases protected by the unbroken symmetry. The critical point between these two, if exists, should exhibit unconventional behaviours both in the bulk and at the boundary.

## Acknowledgments

We thank F. Assaad for illuminating discussions. RM and CW acknowledge support from the Natural Sciences and Engineering Research Council of Canada (NSERC) through a Discovery Grant. Research at Perimeter Institute is supported in part by the Government of Canada through the Department of Innovation, Science and Industry Canada and by the Province of Ontario through the Ministry of Colleges and Universities.

### A. Boundary condition of the gauge field for a disc

In the main text, we have shown that the gauge field $a_\mu$ should be taken to have a Dirichlet boundary condition (up to a gauge transformation). This means that the field strength vanishes and the gauge field does not contain any physical degree of freedom at the boundary. However, though there is always a well defined gauge transformation that sets $a|_{z=0} = 0$ if the system is a half infinite plane, this is generically not true if the system is, say, a disc.

In fact, there is a physical difference between a half infinite plane and a disc. For the former, the only gauge invariant objects that can be constructed from the gauge field $a_\mu$ on the boundary are just the field strengths (and their composites), which vanish. So in this case the boundary gauge field contains no dynamical degree of freedom and can be gauged away, which is the physical reason behind the Dirichlet boundary condition. However, for a disc, besides the field strengths, there is another gauge invariant operator made of $a_\mu$, which is the Wilson loop. In fact, the Wilson loop $\int d\boldsymbol{l} \cdot \boldsymbol{a}$ along the edge counts the physical $U(1)$ charge in the bulk, which cannot be gauged away. It is useful to separate $a_\mu$ into two parts, the part that depends on the spacetime coordinates and contributes to the field strengths, and the part that is independent of the space time coordinates and contributes to the Wilson loop. On a disc, although the former part can still be gauged away since the field strengths vanish, the latter part is a dynamical degree of freedom, so the boundary condition is not really a Dirichlet one.

So how should we think of this boundary condition? Because the spacetime independent part of $a_\mu$ couples to the total gauge charge and current on the boundary, integrating it out in the path integral yields a constraint that the total gauge charge and current vanish on the boundary. Therefore, this boundary condition can be viewed as a Dirichlet boundary condition of $a_\mu$ together with the constraint that the boundary hosts a vanishing gauge charge and current. In contrast, this constraint is absent on a half infinite plane.

### B. Details on emergent anomaly

In this Appendix we review some of the formal aspects of the emergent anomalies in DQCP. A more detailed review of the mathematical backgrounds can be

found in, for example, Ref. [26].

The maximum symmetry of the DQCP is an $O(5)^T$ symmetry, where the superscript "T" means that the improper $\mathbb{Z}_2$ rotation of $O(5)$ should always be accompanied by a time reversal. The $SO(3) \times SO(2) \times \mathbb{Z}_2^T$ of our interest is a subgroup of $O(5)^T$. If we couple the DQCP to a background $O(5)$ gauge field, the partition function is not gauge invariant unless we introduce a bulk topological term living in one extra dimension [54]:

$$S_{bulk}^{O(5)^T} = i\pi \int d^4x\, w_4^{O(5)}, \qquad (B1)$$

where $w_i^{O(5)}$ is the $i$-th Stiefel-Whitney class of the $O(5)$ gauge bundle, with a constraint that $w_1^{O(5)} = w_1^{TM}$ (mod 2) to capture the locking between improper rotation of $O(5)$ and time reversal, where $w_i^{TM}$ is the $i$-th Stiefel-Whitney class of the tangent bundle of the spacetime manifold where the bulk lives. This is a nontrivial t'Hooft anomaly. One consequence of this anomaly is that the system cannot have a boundary if the symmetry is strictly an on-site $O(5)^T$, because the theory can only be defined on the boundary of a higher dimensional bulk and it cannot further have its own boundary. Breaking the symmetry to $SO(3) \times O(2)$ does not help as the anomaly remains nontrivial:

$$S_{bulk}^{SO(3)\times O(2)} = i\pi \int w_2^{SO(3)} \cup w_2^{O(2)}, \qquad (B2)$$

where $\cup$ is the cup product.

The anomaly can be trivialized if we enlarge the symmetry by introducing physical fields that transform as fractional (projective) representations of the original symmetry group. In our case, we are introducing physical fermions that carry half-integer $SO(3)$ spins, odd electric charges and Kramers degeneracy (recall that in our example the superconductor order parameter carries even electric charges). At the level of gauge bundles, the existence of such physical fermions puts the following constraint (cocycle condition) on the Stiefel-Whitney classes, when integrated over any 2-cycle $C_2$:

$$\int_{C_2} [w_2^{SO(3)} + w_2^{TM} + w_2^{O(2)}] = 0 \mod 2, \qquad (B3)$$

The faithful global symmetry is sometimes called $U(2)_c^T$ (the subscript $c$ denotes the fermionic nature of the fundamental operators and the superscript $T$ implies that the fermions are Kramers doublets). Substituting this constraint into Eq. (B2) we get

$$\int (w_1^{TM})^2 \cup w_2^{O(2)} = \int Sq^1(Sq^1 w_2^{O(2)}) = 0, (B4)$$

where the first equality follows from the Wu formula. The anomaly therefore vanishes with the physical fermions we introduce and the system is then allowed to have a consistent edge theory. However, as

the fermions remain gapped in the bulk, the effective symmetry at low energies in the bulk is still $SO(3) \times SO(2) \times \mathbb{Z}_2^T$ and the *emergent* anomaly Eq. (B2) still governs the low-energy response in the bulk, in the sense that a flux quantum in the $SO(2)$ gauge field still traps a Kramers doublet half-integer spin of $SO(3)$, as discussed in the main text.

The anomaly can also be trivialized by other ways of extending the symmetry group. For example, if we introduce physical bosons that carry half-integer SO(3) spins, even electric charge and Kramers degeneracy, the global symmetry becomes $SU(2) \times U(1) \times \mathbb{Z}_2^T$. The cocycle condition is now $w_2^{SO(3)} + (w_1^{TM})^2 = 0$ (mod 2) and the anomaly vanishes. In this case the edge theory is almost identical to the one discussed in the main text, with a boundary Luttinger liquid $(\theta, \phi)$ coupled to the bulk $CP^1$ theory with ordinary boundary condition. The only difference is that the fundamental operators in the edge Luttinger liquid are now bosonic, including $e^{i\theta}$ and $e^{2i\phi}$ (instead of $e^{i(\theta\pm\phi)}$ in the fermionic case).

Note that not all symmetry extensions trivialize the anomaly. For example, if we use $U(2)$ with bosonic Kramers singlet (doublet) fundamental particles, the anomaly becomes $i\pi \int w_2^{SO(2)} \cup w_2^{SO(2)}$ ($i\pi \int (w_2^{SO(2)} + (w_1^{TM})^2) \cup w_2^{SO(2)}$) and is not trivial.

We now return to the fully $O(5)^T$ symmetric anomaly Eq. (B1) and show that it cannot be trivialized. For this purpose, it suffices to consider orientable spacetime manifold for the bulk, such that the anomaly in Eq. (B1) reduces to $i\pi \int d^4x\, w_4^{SO(5)}$. It is known that if we introduce physical fermions in the spinor representation of the $SO(5)$ (so that the faithful symmetry is $Spin(5)_c$), the DQCP can be realized if we put the fermions in half-filled Landau levels [52]. At the anomaly level the existence of the physical $Spin(5)$ fermions imposes the cocycle constraint

$$w_2^{SO(5)} = w_2^{TM} \mod 2. \qquad (B5)$$

We then use the relation (true for $SO(N)$ bundles on a $4d$ orientable manifolds)

$$w_4^{SO(5)} = \frac{1}{2}p_1 - \frac{1}{2}\mathcal{P}(w_2^{SO(5)}) \mod 2, \qquad (B6)$$

where $\mathcal{P}$ is the Pontryagin square and $p_1$ is the first Pontryagin number

$$p_1 = \frac{1}{8\pi^2} \int \mathrm{tr}_{SO(5)} F \wedge F. \qquad (B7)$$

We can now substitute the cocycle condition Eq. (B5) into the above equations and use the relation for the signature $\sigma$ of a $4d$ manifold (with Riemann curvature tensor $R$):

$$\frac{\sigma}{2} = \frac{1}{2}\mathcal{P}(w_2^{TM}) = -\frac{1}{48\pi^2} \int R \wedge R \mod 2. \qquad (B8)$$

We conclude that the anomaly becomes the following gauge and gravity theta terms:

$$S_{bulk}^{Spin(5)_c} = \frac{1}{16\pi} \int \mathrm{tr}_{SO(5)} F \wedge F + \frac{1}{48\pi} \int R \wedge R. \quad (B9)$$

The above theta term is nontrivial for $Spin(5)_c$ symmetry as long as time-reversal (or any orientation-reversing) symmetry is preserved. This means that for the $Spin(5)_c$ system to have a physical boundary, either the $Spin(5)$ or time-reversal symmetry must be explicitly broken on the boundary in order to have a consistent boundary theory. In our main example the $Spin(5)$ symmetry is broken to $(U(1) \times SU(2))/Z_2 = U(2)$ while time-reversal symmetry is preserved. In the half-filled Landau level [52] the $Spin(5)$ symmetry is kept but the anti-unitary particle hole symmetry (playing the role of time-reversal) is broken on the boundary due to the nature of Landau levels.

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
