# Peer review of "Edge physics at the deconfined transition between a quantum spin Hall insulator and a superconductor"

_SciPost Physics_

## Round 1 · Referee Report · Anonymous (Referee 1) · 2021-11-24

Strengths

1- Good work 2- Well-motivated 3- Generally well-written

Weaknesses

A few points need clarification, see below.

Report

This paper extends the study of gapless SPTs by giving a detailed analysis of a parton description of the deconfined transition from spin-Hall to superconductor. The authors point out several anomalous, or at least strange, features of the edge theory.
In particular, they argue that this is an example of "symmetry-enforced gaplessness". An important role is played by the SU(2) spin rotation symmetry.
They make some quantitative statements for a generalization where SU(2) spins are replaced by SU(N) spins.
It is a good paper and should be published.
I have a few improvements to suggest.

-- Among the list of studies of critical points of SPTs and their consequences for the boundary physics, the reference list should include
https://arxiv.org/abs/1206.1332
where the fact that the single-electron gap remains open in the bulk also plays an important role.

I was confused by several points in the argument of section IV about the anomaly:

-- First, I think the authors should be clearer about the context for the discussion of section IV. If I understand correctly, they are sitting at the critical point, where the bulk gap vanishes, but the bulk fermion gap does not.

-- Is it obvious that an argument based on adiabatic flux threading always still works even in the absence of a gap to neutral excitations? I think this merits some attention.

-- The authors argue for the gapless edge modes via a flux-threading argument on the disk, which seems to imply a contradiction in the absence of gapless edge modes. Why doesn't a similar contradiction arise if instead of on a disk, one inserts flux in some region of a closed spatial manifold?
Then there is no gapless boundary to save the day.

-- What exactly is "emergent anomaly" more generally? It would be nice to have a clear definition of this term.

-- Should we expect that the jump in the dimension of the boundary operator that the authors observe is smoothed out at finite $N$?

-- The anomalous nature of the gapless edge rests on the impossibility of a fully SU(2)-invariant topological insulator.
At several points the authors assert the well-known-ness of this impossibility, but do not give any argument for it.
Is there actually an argument for this statement that shows the impossibility not just for free fermion models? Even for free fermions, I only know this statement from trying and failing.

-- I didn't understand at all the meaning of the phrase "emergent anomaly remains a non-trivial $\theta$-term".

-- A request for clarification: I didn't understand the discussion about the boundary breaking SO(5) and its connection to anomalies. It seems to me that the critical theory on the edge will break SO(5) if the SO(5)-breaking operators are not irrelevant at the fixed point. How do the existence of various anomalies bear on this issue?

Requested changes

Please see the report.

smaller things:
-- an extra "the" in "consequences of the the gapless edge states"

-- "truely" --> "truly"

-- on page 8, "several measurable consequence" --> "several measurable consequences"

-- footnote 6: "to what extend" --> "to what extent"

  • validity: -
  • significance: -
  • originality: -
  • clarity: -
  • formatting: -
  • grammar: -

Author:  Liujun Zou  on 2022-04-06  [id 2363]

(in reply to Report 1 on 2021-11-24)
Category:
answer to question

The comments and questions from the referee are highly appreciated. Here is a reply regarding the points raised by the report (the requested changes from the referee have also been made).

  1. Indeed, at the critical point we discussed, the single electron gap remains open in the bulk. The effective low-energy symmetry is therefore the quotient of the full symmetry group by the part that acts non-trivially only on the gapped degrees of freedom (the pi-flux of the global U(1), or fermion parity, in our example). A theory with an ``emergent anomaly" denotes the case where the low-energy symmetry is anomalous, although the full symmetry group is anomaly free. The emergent anomaly could lead to constraints on IR physics, such as the existence of edge modes which are charged under the gapped symmetry of the bulk (e.g. the gapless fermion modes in our paper). We have clarified these points in the revised manuscript.

  2. The meaning of adiabatic insertion means that the flux is inserted so slowly, such that the bulk fermions will not be excited. Since the bulk fermions are always gapped, this adiabatic insertion is well defined. We have clarified this in Sec. IV.

  3. On the disk, we can consider an insertion of $\pi$-flux. The emergent anomaly of the low-energy symmetry demands such a $\pi$-flux carry half-integral spin, which then results in the existence of gapless fermions on the edge. However, on a closed spatial manifold, the total flux is quantized as integral multiples of $2\pi$. This flux always has an integral spin moment, thus the absence of edge modes leads to no contradiction.

  4. See response to 1.

  5. In the paper based on a calculation at infinite $N$, we showed that at the criticality there is a universal jump in the electron scaling dimension at the edge. At finite $N$, one may ask (a) whether the edge Luttinger liquid still decouples with the critical bulk; (b) whether the value of the jump remains the same (or universal). Regarding (a), at small N the most dangerous coupling is between the edge Cooper pair and the bulk monopole operator. At least in the parameter regime where the edge Cooper pair is heavy enough, this coupling is irrelevant, hence the edge fermions remain decoupled with other degrees of freedom. Then there is still a jump, whose value is determined by the boundary scaling dimension of the CP$^1$ field, which may deviate from 1 at finite $N$, but remains universal. This is a manifestation of electron fractionalization at the transition. These are discussed around Eqs. (27, 28).

  6. We have added an argument for this: suppose it is possible, adiabatically threading a pi-flux would result in a spin-1/2, which is doubly degenerate. This change of degeneracy is well defined if the system is gapped to start with, and it contradicts the adiabaticity of the flux insertion.

  7. Thanks for this question, and we have clarified and expanded this section, and we have also added a new appendix for more details.

  8. The reason that the anomaly can imply that the $SO(5)$ cannot emerge on the boundary is as follows. The bulk has an anomaly associated with the $SO(5)$ symmetry, but the vacuum does not. As their interface, the anomaly cannot be well defined on the boundary. For this to happen, the $SO(5)$ symmetry needs to be broken on the boundary. In terms of RG flow, this means that the boundary is simply incompatible with an $SO(5)$ symmetry, so we cannot talk about whether $SO(5)$-breaking perturbations are irrelevant or not.

---

## Round 1 · Referee Report · Anonymous (Referee 2) · 2021-12-15

Strengths

  1. Novel results on robust edge state in the absence of a bulk energy gap

  2. Clear presentation in general

  3. Technically sound

Weaknesses

1. The discussion of the emergent anomaly may need further clarification

Report

The authors studied the edge physics at the continuous phase transition between the quantum spin Hall (QSH) insulator that breaks the SO(3) spin rotation symmetry and the s-wave superconductor that breaks charge U(1) symmetry. This transition was captured by a deconfined quantum critical point in the 2+1d bulk. Using a large-N analysis and anomaly-based arguments, the authors provided strong evidence that non-trivial localized edge states not only exist in the QSH phase but also at the critical point. The authors also showed that the fermion scaling dimension at the edge experiences a universal jump as one approaches the critical point from the QSH side of the phase diagram. I believe that this is a nicely written paper with novel results and, hence, should be published once the following comments and questions are addressed:

1. On page 4, the authors made a side remark that if one did not include the helical edge fermions in the beginning, the \phi field would still form a Luttinger liquid on the edge. Can the authors clarify if the two theories with and without the explicit inclusion of the edge helical fermions are the same theory or not? Naively, they should be different because the theory with explicit inclusion of the helical edge fermions is intrinsically a theory of a fermion system. In contrast, the theory without the helical edge fermions is purely bosonic. Is this correct?

2. The two theories with and without the explicit inclusion of the edge helical fermions can share the same edge description Eq. (13) to begin with. Is it correct that all the results (except the ones directly related to the electron operator) are essentially the same in the two theories? More generally, could the authors comment on how important it is to have local fermions in this problem?

3. In the QSH phase, the wavefunctions of the helical edge fermions are exponentially localized near the edge of the system. The authors showed that the helical edge fermions are decoupled from bulk critical fluctuations under RG flow at the critical point. Does it mean that the helical edge fermions are still exponentially localized near the edge when the bulk is critical? Or would the irrelevant couplings to the bulk change result in a power-law localization of the edge fermions instead?

4. I think it would be helpful for the readers if the authors could expand the discussion in Sec. IV. In particular, the concept of the emergent anomaly itself may deserve a more general introduction. Also, I hope the authors can clarify what the "theta term" referred to in the sentence “At the anomaly level this is because the emergent SO(5) anomaly remains a nontrivial \theta-term…”. Also, regarding the last sentence “So the boundary gapless fermions can only trivialize the bulk SO(3) × U(1) anomaly, but not the larger SO(5) anomaly, unless time-reversal symmetry is broken with nonzero SO(5) and thermal Hall conductance”, can the author clarify why “a nonzero SO(5) and thermal Hall conductance” is important? In other words, why is time-reversal symmetry breaking alone is not enough?

Requested changes

see report

  • validity: top
  • significance: high
  • originality: top
  • clarity: high
  • formatting: perfect
  • grammar: -

Author:  Liujun Zou  on 2022-04-06  [id 2364]

(in reply to Report 2 on 2021-12-15)
Category:
answer to question

The comments and questions from the referee are highly appreciated. Here is a reply regarding the points raised by the report.

  1. We find that the original discussion of the boundary condition is rather confusing, so we have changed it in the revised manuscript. In particular, the discussion of $\varphi$ is removed. We have also added an Appendix A, where we show how this should be modified when the system is a disc. In particular, for a disc there is one more boundary degree of freedom compared to the case of half infinite plane, which is just the Wilson loop.

  2. See response to 1.

  3. Since the coupling is irrelevant, this edge mode should still be exponentially localized in the presence of the gapless bulk. In other words, the two-point correlation function of the fermionic operator decays exponentially in the direction perpendicular to the edge.

  4. Thanks for this suggestion. Referee 1 had the same suggestion, so we have revised and expanded that section.

---

## Editorial Decision

resubmitted